# Inverted U-Shaped Relationship between Obesity Parameters and Bone Mineral Density in Korean Adolescents

**DOI:** 10.3390/jcm12185869

**Published:** 2023-09-09

**Authors:** Jongseok Lee, Insang Yoon, Hwajung Cha, Ho-Jung Kim, Ohk-Hyun Ryu

**Affiliations:** 1School of Artificial Intelligence Convergence, Hallym University, Chuncheon 24253, Republic of Korea; pny235711@gmail.com (I.Y.); nataliac@kakao.com (H.C.); hojungkim@hallym.ac.kr (H.-J.K.); 2Division of Endocrinology and Metabolism, Department of Internal Medicine, Chuncheon Sacred Heart Hospital, Chuncheon 24253, Republic of Korea

**Keywords:** adolescent, obesity, body composition, bone development, bone mineral density, cross-sectional studies

## Abstract

As the association between obesity and bone health remains controversial in children and adolescents, we investigate the effects of obesity parameters on bone mineral density (BMD) in 2060 Korean adolescents who participated in the 2008–2011 Korea National Health and Nutrition Examination Survey (KNHANES). Multiple regression analysis and analysis of covariance (ANCOVA) were conducted to examine both the linear and non-linear associations between total-body-less-head (TBLH) BMD and four obesity parameters: body mass index, waist circumference, waist-to-height ratio, and total-body fat mass (FM). In a multiple linear regression analysis adjusted for age, menarcheal status (in females only), and total-body lean mass, there was no significant linear association between obesity parameters and TBLH BMD, except for total-body FM in males. However, upon adding a second-order polynomial term for each obesity parameter, a significant quadratic relationship between all obesity parameters and TBLH BMD was observed, with the corresponding quadratic term being negative. The results of ANCOVA also revealed an inverted U-shaped relationship between each obesity parameter and TBLH BMD. Our findings suggest the existence of an optimal range of obesity parameters for developing or maintaining optimal bone health in Korean Adolescents. Deviation from this range, in either direction (being underweight or having obesity), may compromise bone health.

## 1. Introduction

Both obesity and osteoporosis are serious global health concerns that are experiencing a rise in prevalence, leading to substantial impacts on mortality and morbidity [1,2,3,4,5]. Not only in adults but also in children and adolescents, the worldwide prevalence of obesity has been steadily increasing for several decades [6,7,8]. Childhood obesity tends to persist into adulthood and is linked to cardo-metabolic and psychosocial comorbidity as well as premature mortality [9,10,11]. Being overweight or obese is associated with the development of various musculoskeletal problems, ranging from general discomfort and pain in the joints and muscles to more severe injuries such as fractures, even during childhood [12,13]. The association between obesity and musculoskeletal disorders highlights the importance of addressing body composition and promoting optimal bone health from an early age to minimize the risk of these complications.

Adolescence is characterized by dramatic changes in body composition that significantly influence bone health [14]. It is well-known that childhood and adolescence are crucial periods for skeletal development and maturation, with approximately half of peak bone mass being attained in childhood and nearly all of adult bone mass being achieved by the end of adolescence [15]; therefore, the amount of bone mass accrued during these periods is a major determinant of the adult’s risk for osteoporosis [16]. Recognizing the significance of this developmental phase and understanding how aspects of body composition, such as obesity, influence bone mineral accrual is essential for promoting optimal bone health and reducing the risk of fractures later in life. 

Recently, there has been growing concern regarding the impact of obesity or excessive fat mass (FM) on bone development in children and adolescents; however, it remains controversial whether obesity has a beneficial or detrimental effect on the growing skeleton, with conflicting and inconclusive findings from studies. While some studies have reported positive associations between obesity parameters and bone health [17,18,19,20,21,22], others have found negative associations [23,24,25,26,27,28,29]. There is also evidence indicating no significant association between obesity and bone health in children and adolescents [30,31,32,33,34]. Positive associations emphasize the risk of being underweight, whereas negative associations highlight the risk related to being overweight or obese. We tested the hypothesis that both underweight and overweight/obesity can have detrimental impacts on bone health, thus, prompting the exploration of potential non-linear relationships between obesity and bone health.

The objective of this study was to investigate the impact of obesity on bone development in Korean adolescents aged 10–19 years. Obesity was assessed using four indicators: body mass index (BMI), waist circumference (WC), waist-to-height ratio (WHtR), and total-body FM. We examined the linear relationship between each obesity parameter and total-body-less-head (TBLH) bone mineral density (BMD), separately for males and females, while controlling for age, menarcheal status (in females only), and total-body lean mass (LM). Moreover, we explored the possibility of a quadratic relationship between each obesity parameter and TBLH BMD. By considering the potential non-linear associations, our study contributes to the current understanding in this field.

## 2. Materials and Methods

### 2.1. Study Population

The study population consisted of 2060 Korean adolescents, including 1114 males and 946 females, aged 10–19 years, who were selected from the Korea National Health and Nutrition Examination Survey (KNHANES) conducted between 2008 and 2011. KNHANES is a nationwide, population-based, cross-sectional survey conducted by the Korea Disease Control and Prevention Agency (KDCA) since 1998. It utilizes a stratified, multistage, clustered probability sampling method to select a representative sample of the non-institutionalized civilian Korean population. The survey was approved by the Institutional Review Board of the KDCA (approval codes: 2008-04EXP-01-C, 2009-01CON-03-2C, 2010-02CON-21-C, 2011-02CON-06-C), and all participants provided written informed consent. 

For the present study, we included teenagers who completed body composition and bone examinations. Out of 5105 teenagers who participated in the 2008–2011 KNHANES, we excluded those who did not undergo body composition and bone examinations, as well as those who had missing physical measurements or who did not respond to self-reported questionnaires about menstruation. The total number of participants in the analysis was 2060.

### 2.2. Measurements

Body composition and bone parameters were measured by DXA using a DISCOVERY-W fan beam densitometer (Hologic Inc., Waltham, MA, USA) according to standard procedures. The total body mass is composed of LM, FM, and bone mass. In this study, TBLH BMD was used to evaluate bone health. TBLH BMD was chosen instead of total-body BMD to minimize the impact of the skull and teeth on BMD measurements, which can lead to inaccurate results [35]. This is particularly important in studies that involve children and adolescents, as their heads are proportionally larger than those of adults, and differences in head size can significantly affect BMD measurements [36]. TBLH measurements are widely used for relative comparisons of bone density and are particularly suitable for populations with obesity or underweight, where TBLH measurements provide more accurate results than total-body measurements [37].

Anthropometric variables such as body weight, height, and WC were measured in this study. Participants were instructed to wear light clothing and remove their shoes during weight measurements, which were recorded to the nearest 0.1 kg. Height measurements were taken to the nearest 0.1 cm using a wall-mounted stadiometer. WC was measured to the nearest 0.1 cm at the midpoint between the lower border of the rib cage and the iliac crest. BMI was calculated by dividing the weight in kilograms by the square of the height in meters. WHtR was calculated by dividing WC in centimeters by the height in centimeters.

Obesity was assessed using four measures of obesity: BMI, WC, WHtR, and total-body FM. Based on each of these obesity parameters, subjects were categorized into four percentile groups: underweight (<5th percentile), normal-weight (5th–85th percentile), overweight (86th–94th percentile), and obesity (≥95th percentile). The BMI percentile groups were stratified based on the 2007 Korea National Growth Charts [38], considering gender and age (Appendix A). This stratification followed the recommendations set forth by the United States Centers for Disease Control and Prevention (CDC) Growth Reference for children and adolescents aged 2 to 20 years [39]. The percentile groups of WC, WHtR, and total-body FM were determined using the observed sample distributions in the study, separately for males and females (Appendix A).

Pubertal development was assessed through self-reported questionnaires, which included inquiries about the presence or absence of menstruation and the onset of menarche. These assessments were used to determine the pubertal stage of female participants.

### 2.3. Statistical Analysis

The study population characteristics were presented as means and standard deviations (SD) for the study variables and compared between genders using a t-test for two independent samples. To evaluate the differences in mean values of TBLH BMD among the four percentile groups of each obesity parameter (BMI, WC, WHtR, and total-body FM) separately for males and females, one-way analysis of variance (ANOVA) was performed. 

Multiple regression analysis was conducted to examine the linear relationship between each obesity parameter and TBLH BMD while controlling for age, menarcheal status (for females only), and total-body LM. However, if there is a curved relationship between the obesity parameter and TBLH BMD, then a linear model may not be the best fit. Therefore, the possibility of a curvilinear relationship between each obesity parameter and TBLH BMD was explored by incorporating a second-order polynomial term for each obesity parameter (BMI^2^, WC^2^, WHtR^2^, or FM^2^) into the regression model. For example, the statistical model used to examine the quadratic effect of BMI on TBLH BMD, characterized by a single turning point, is as follows:TBLH BMD = *β*_0_ + (*β*_1_ × Age) + (*β*_2_ × LM) + (*β*_3_ × BMI) + (*β*_4_ × BMI^2^) + *ε*(1)
where *β*_0_ = intercept

*β*_1_ × Age = linear effect of Age*β*_2_ × LM = linear effect of total-body LM*β*_3_ × BMI = linear effect of BMI*β*_4_ × BMI^2^ = quadratic effect of BMI

This modeling approach allows us to directly investigate whether a quadratic effect of BMI on TBLH BMD exists, with the coefficient *β*_4_ and its statistical significance. In terms of interpretation, a positive quadratic coefficient indicates a ∪-shaped curve, whereas a negative coefficient indicates a ∩-shaped curve. To assess for a significant quadratic trend in the means of TBLH BMD across percentile groups for each obesity parameter, while adjusting for the covariates, analysis of covariance (ANCOVA) was used, and the results were reported graphically.

The IBM SPSS Statistics for Windows (Version 27; IBM Corp., Armonk, NY, USA) was used for data analysis. All statistical tests were two-sided, and *p*-values less than 0.05 were considered statistically significant.

## 3. Results

### 3.1. Characteristics of Study Population

Table 1 presents the characteristics of the study population, including the means of the study variables stratified by gender. The study comprised 2060 adolescent subjects, with 1114 males and 946 females, and mean ages of 14.17 and 14.28 years, respectively. Except for age and percent bone mass means, all other variables exhibited statistically significant differences between males and females. Females exhibited significantly higher values for two FM-related variables (FM and percent FM), while males had significantly higher values for other obesity, body composition, and TBLH bone parameters.

### 3.2. Age-Related Changes in TBLH BMD, Total-Body LM, and Total-Body FM

In this study, we observed age-related changes in mean values of TBLH BMD, total-body LM, and total-body FM. Figure 1 depicts the age-related changes in the means and 95% confidence intervals (CI) of TBLH BMD, total-body LM, and total-body FM, separately by gender (refer to Appendix A for sample distribution according to age groups, separately for males and females). The findings indicate that the age-related changes in mean TBLH BMD exhibited a pattern similar to the changes in mean total-body LM, rather than the changes in mean total-body FM. The mean values of TBLH BMD increased with age in both males and females but the rate of increase was more prominent in males, particularly after the age of 14. Consequently, the mean TBLH BMD values between males and females were not significantly different from ages 10 to 13 but males had significantly higher values than females from ages 14 to 19 (Appendix A). Regarding the mean values of total-body LM, males showed a continuous increase from ages 10 to 19, while females experienced relatively stable levels after the age of 14. On the other hand, the mean values of total-body FM in females continued to increase from ages 10 to 19, while in males, there was no significant increase after the age of 12. As a result, females had significantly higher mean values of total-body FM than males from ages 13 to 19 (Appendix A).

### 3.3. Crude Relationship between Obesity Parameters and TBLH BMD

The unadjusted relationship between obesity parameters (BMI, WC, WHtR, and total-body FM) and TBLH BMD was examined using correlation and simple regression analysis by gender, as shown in Figure 2. Each obesity parameter exhibited a positive linear association with TBLH BMD in both males and females (all *p*-values < 0.001). The proportion of variance in TBLH BMD explained by the different obesity parameters as follows: BMI explained 22.4% and 28.5% in males and females, respectively; WC explained 27.6% and 29.9%; WHtR explained 1.3% and 10.2%; and total-body FM explained 6.0% and 25.6%. In addition, the mean values of TBLH BMD were compared among four percentile groups (<5th percentile, underweight; 5th–85th percentile, normal-weight; 86th–94th percentile, overweight; ≥95th percentile, obesity) stratified on the basis of each obesity parameter (refer to Appendix A for the mean comparison of age among four percentile groups). Except for WHtR in males, there was a significant linear trend for the means of TBLH BMD to increase across the percentile groups of each obesity parameter in both males and females (Appendix A).

### 3.4. Linear Relationship between Obesity Parameters and TBLH BMD

Table 2 presents the results of multiple regression analysis separately by gender, adjusted for age, menarcheal status (in females only), and total-body lean mass. There was no significant linear association between TBLH BMD and the obesity parameters, except for total-body FM in males, which showed a significant negative association with TBLH BMD. In all models, age and total-body LM were positively related to TBLH BMD, with total-body LM demonstrating the highest explanatory power in terms of the largest partial R. Furthermore, females who had started menstruation had higher TBLH BMD than those who had not, assuming all other conditions were the same. The models for males explained 73.5% to 73.8% of the variance in TBLH BMD, while the models for females accounted for about 60.8% of the variance.

### 3.5. Non-Linear Relationship between Obesity Parameters and TBLH BMD

To explore the possibility of a non-linear relationship between the obesity parameters and TBLH BMD, a second-order polynomial term for each of the obesity parameters (BMI^2^, WC^2^, WHtR^2^, or FM^2^) was added to the regression model in Table 2. The results, as presented in Table 3, revealed a significant quadratic relationship between all obesity parameters and TBLH BMD in both genders, with the corresponding quadratic term being negative, indicating an inverted U-shaped curve (refer to Appendix A for graphical representation). Moreover, all coefficients of the independent variables were statistically significant, and the inclusion of the quadratic terms enhanced the explanatory power of the models. Specifically, the models for males explained 74.0% to 74.3% of the variance in TBLH BMD, while the models for females accounted for 61.6% to 62.1% of the variance.

Figure 3 illustrates the inverted U-shaped relationship between each obesity parameter and TBLH BMD, separately for males and females, after adjusting for age, menarcheal status (for females), and total-body LM. The figure displays the mean and 95% CI of TBLH BMD among the four percentile groups based on each obesity parameter. It can be observed that there was a significant quadratic trend for the means of TBLH BMD to increase across the percentile groups of each obesity parameter in both males and females (all *p*-values for the quadratic trend < 0.005). The mean values of TBLH BMD increased from the underweight group to the normal-weight or overweight group but subsequently decreased as the percentile groups progressed towards obesity.

## 4. Discussion

In this study, we found that the relationship between obesity parameters and TBLH BMD in Korean adolescents did not follow a linear pattern but rather an inverted U-shaped curve, after controlling for age, menarcheal status (in females only), and total-body LM. This indicates that obesity parameters are positively related to TBLH BMD until reaching a healthy weight from being underweight; however, if obesity parameters exceed a certain threshold, such as the overweight or obese state, they are negatively related to TBLH BMD.

Adolescence is a critical period for achieving peak bone mass, which has long-term implications for bone health later in life. Extensive research has been conducted to investigate the influence of body composition and obesity on bone health during this crucial growth phase. Most studies have emphasized the role of LM as a strong predictor of bone mass and density, demonstrating a positive association in children and adolescents [20,21,22,32,33,34], which is consistent with our findings; however, the impact of obesity or excessive FM on bone remains controversial. Some studies have reported positive associations between obesity parameters and bone health [17,18,19,20,21,22], suggesting that being overweight or obese may have beneficial effects on bone. These findings also highlight that being underweight is widely recognized as a risk factor for fractures [40]. Conversely, other studies have found negative associations between these variables [23,24,25,26,27,28,29], implying that obesity could have detrimental effects on bone. Furthermore, there is evidence to indicate no significant association [30,31,32,33,34], suggesting that the relationship might be indirect and mediated by other factors, such as the contribution of lean mass.

These inconsistencies in the findings may be largely attributed to the confounding effects of growth, maturation, and body composition. In this study, we initially observed a significant positive association between each of the obesity parameters (BMI, WC, WHtR, and total-body FM) and TBLH BMD in both males and females when considering the unadjusted regression analysis (Figure 2). However, after controlling for age, menarcheal status (in females only), and total-body LM, these associations were no longer observed and became non-significant (Table 2). The exception was total-body FM in males, which showed a significant negative association with TBLH BMD.

Our findings are supported by previous studies that also adjusted for total-body LM and reported no significant associations between obesity parameters and bone health. For instance, Hage et al. [31] found no significant differences in total-body BMD among different BMI groups (obese, overweight, and normal) in adolescent girls after adjustment for LM. Similarly, Valdimarsson et al. [32] and Witzke et al. [33] reported no significant association between total-body FM and total-body BMD in females. Gracia-Marco et al. [34] reported a significant negative association between FM and total-body BMD in males but not in females, which aligns with our results. Additionally, studies investigating the relationship between FM and femoral neck BMD also reported non-significant associations [22,32,34]. These findings further support the notion that the relationship between obesity parameters and bone health may be influenced by other factors, such as LM.

Indeed, there is growing evidence to suggest that excessive fat mass, rather than obesity per se, may have a negative impact on bone health in children and adolescents. Several studies have reported associations of abdominal fat with decreased bone formation rate and lower bone quality compared to resorption. Additionally, higher amounts of FM have been found to be a risk factor for fractures in adolescence [41,42]. This negative effect is thought to be due to the pro-inflammatory effects of adipose tissue on bone cells, leading to increased bone resorption and decreased bone formation. Excessive adipose tissue in obese individuals is known to secrete various inflammatory cytokines, such as interleukin-6 and tumor necrosis factor-alpha [43]. The increased secretion of these cytokines, particularly from visceral fat, may contribute to a decrease in bone density by promoting greater bone resorption [44,45]. Furthermore, insulin resistance, which is commonly associated with obesity, has been found to be negatively associated with BMD in obese adolescents [46]. Insulin resistance can disrupt normal bone metabolism and lead to impaired bone formation. These findings highlight the complex interactions between adipose tissue, inflammation, and bone health. Excessive FM, especially when accompanied by systemic inflammation and insulin resistance, may contribute to compromised bone health in children and adolescents.

Both underweight and overweight/obesity conditions can have detrimental impacts on bone health, indicating the need to investigate the dual effects of obesity status on BMD. To explore the possibility of a non-linear relationship between the obesity parameters and TBLH BMD, we added a second-order polynomial term for each of the obesity parameters (BMI^2^, WC^2^, WHtR^2^, or FM^2^) to the regression model. The results demonstrated a significant quadratic relationship between all obesity parameters and TBLH BMD in both males and females (Table 3). The negative coefficient of the quadratic term indicates an inverted U-shaped curve, as illustrated in Figure 3. The mean values of TBLH BMD showed an upward trend from the underweight group to the normal weight or overweight group; however, as the percentile groups progressed towards obesity, a subsequent decline in TBLH BMD was observed. This suggests that there may be an optimal range of obesity parameters for maintaining optimal bone health, with deviations from this range in either direction (being underweight or having obesity) potentially leading to compromised bone health.

Recent studies have reported non-linear associations between obesity and bone health, which further support our findings. For instance, two studies involving Chinese children aged 0–5 years [47] and Spanish children aged 4–18 years [48], respectively, revealed that the positive association between BMI and BMD reached a plateau or attenuated at higher levels of obesity, suggesting that there is a saturation point beyond which increasing BMI does not lead to additional increases in BMD. A similar saturation effect of BMI on BMD was also observed in a cross-sectional study involving US adolescents aged 8–19 years, indicating that maintaining BMI at saturation values may have benefits in mitigating other adverse effects associated with obesity [49]. These studies highlight the importance of considering the non-linear nature of the relationship between obesity and bone health, emphasizing the need to maintain obesity parameters within an optimal range to achieve optimal BMD.

In this study, we observed a significant association between menarcheal status and bone health. After controlling for the covariates including age, we found that post-menarcheal females had higher TBLH BMD compared to pre-menarcheal females (Table 2 and Table 3). This finding suggests that the onset of menarche, which represents the beginning of reproductive maturity in females, may have a positive impact on bone health. Menarche is accompanied by hormonal changes, particularly an increase in estrogen levels. Estrogen plays a crucial role in bone metabolism, promoting the growth and strength of bones [50]. It enhances the lifespan and activity of osteoblasts—the cells responsible for bone formation—while inhibiting the activity of osteoclasts, the cells involved in bone resorption [51]. Therefore, the higher TBLH BMD observed in post-menarcheal females can be attributed to the beneficial effects of estrogen on bone health. The timing and duration of exposure to estrogen during puberty play a vital role in determining peak bone mass. A shorter period of exposure to higher estrogen levels, particularly in late menarcheal girls, is considered a risk factor for decreased BMD and increased susceptibility to osteoporosis later in life.

Our study has some limitations. First, this is a cross-sectional analysis, which does not allow us to identify the causal effects of obesity on BMD. Second, it is important to note that DXA measures areal bone density (in g/cm^2^), not volumetric bone density (in g/cm^3^). DXA bone area measurement captures the length and width of bones but does not directly assess the greater depth or thickness of bones. Therefore, individuals with larger bone sizes may have a greater areal BMD reading, even if their volumetric BMD is the same [52]. Third, we were not able to include variables to assess physical activity, vitamin D levels, nutritional status [53], and other chemical mediators between fat and bone, including inflammatory cytokines and gonadal steroids. Finally, in order to better understand the impact of obesity on bone density, it would be necessary to determine regional fat distribution using computed tomography or magnetic resonance imaging; however, we were unable to differentiate between subcutaneous and visceral fat.

Despite these limitations, our study has several strengths. One notable advantage is the sampling design of the KNHANES, which ensures that our study sample is nationally representative of Korean adolescents, allowing for generalization to the broader Korean adolescent population. Furthermore, we addressed a statistical methodological concern by employing multiple regression modeling to adjust the associations for relevant covariates, including age, menarcheal status, and total-body LM. By considering these factors, we aimed to provide a more accurate assessment of the associations between obesity parameters and BMD. Moreover, it is worth noting that previous studies have predominantly focused on examining the linear relationship of obesity with bone parameters. This emphasis on linearity may be one of the reasons for the seemingly contrasting evidence reported in the literature regarding the relationship between obesity and bone health. By exploring potential non-linear associations, our study adds to the current understanding in this field.

## 5. Conclusions

In conclusion, we investigated the relationship between four obesity parameters and TBLH BMD to examine the impact of obesity on bone health. The analysis revealed a tendency where TBLH BMD increased with increasing the obesity parameters; however, an inverted U-shaped pattern was observed, indicating that TBLH BMD decreased as the obesity parameters reached a certain threshold, such as the overweight or obese state. In other words, optimal BMD was achieved within a specific range of obesity parameters, and deviation from this range, whether towards underweight or obesity, could lead to BMD deterioration. Considering obesity in childhood and adolescence persists into adulthood and produces more harmful effects on health, maintaining a healthy weight in adolescence is an essential aspect of preventing osteoporosis in older adults. To clarify the causal relationship between obesity and bone health, intervention studies aimed at maintaining a healthy weight in the adolescent population with obesity should be conducted.

## Figures and Tables

**Figure 1 jcm-12-05869-f001:**
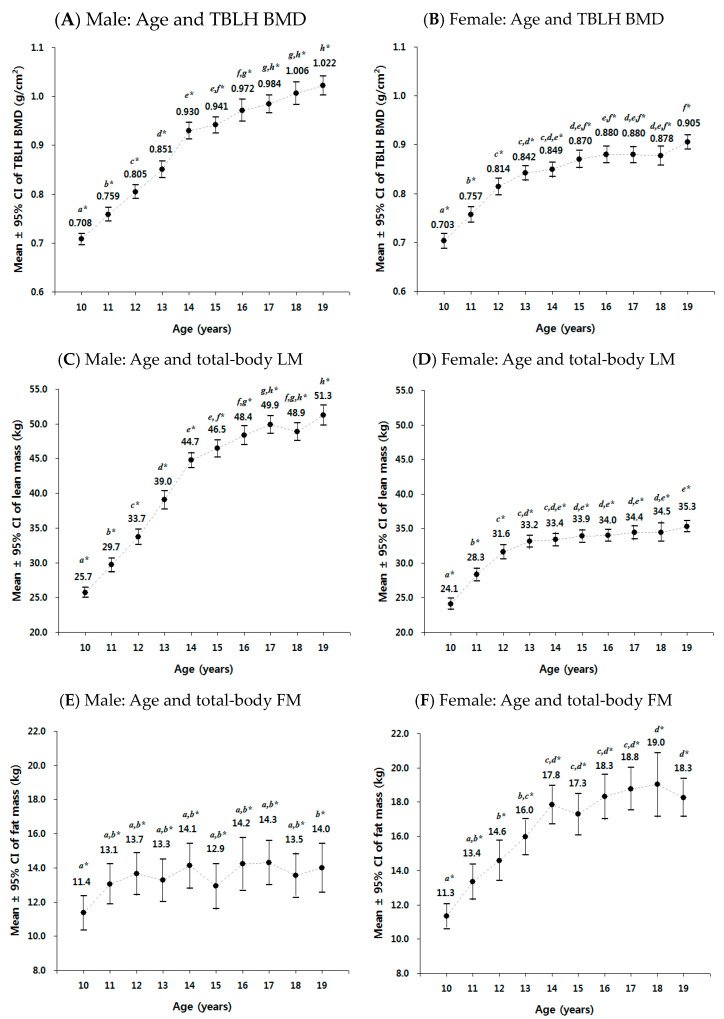
Age-related changes in total-body-less-head (TBLH) bone mineral density (BMD), total-body lean mass (LM), and total-body fat mass (FM) by gender in 2060 Korean adolescents aged 10 to 19 years from the Korea National Health and Nutrition Examination Survey (KNHANES) 2008–2011. Mean ± 95% confidence interval (CI) was calculated by one-way analysis of variances (ANOVA). The figure presented the mean ± 95% confidence interval of TBLH BMD for (**A**) males and (**B**) females, total-body LM for (**C**) males and (**D**) females, and total-body FM for (**E**) males and (**F**) females. * The same letters indicate that the mean difference between groups is not significant (α = 0.05) on pairwise post-hoc comparisons using Bonferroni’s test (equal variance assumed) or the Games–Howell test (equal variance not assumed).

**Figure 2 jcm-12-05869-f002:**
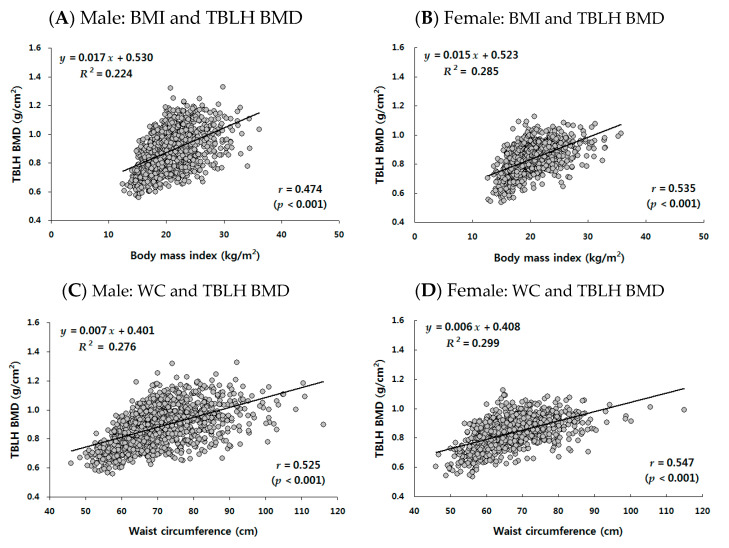
Unadjusted relationships between obesity and total-body-less-head (TBLH) bone mineral density (BMD) by gender in 2060 Korean adolescents aged 10 to 19 years from the Korea National Health and Nutrition Examination Survey (KNHANES) 2008–2011. The figure illustrates the simple regression results showing the associations of TBLH BMD with four obesity parameters: body mass index (BMI) for (**A**) males and (**B**) females, waist circumstance (WC) for (**C**) males and (**D**) females, waist-to-height ratio (WHtR) for (**E**) males and (**F**) females, and total-body fat mass (FM) for (**G**) males and (**H**) females. The black line is the fitted regression line. “*r*” is Pearson’s correlation coefficient.

**Figure 3 jcm-12-05869-f003:**
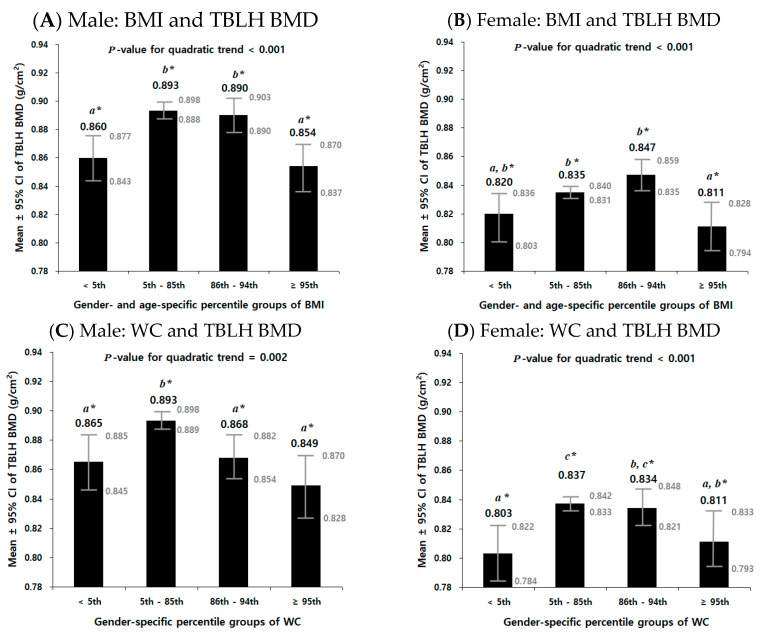
Quadratic relationships between obesity and total-body-less-head bone mineral density (TBLH BMD) adjusted for age, menarcheal status (for females only), and total-body lean mass by gender in 2060 Korean adolescents aged 10 to 19 years from the 2008–2011 Korea National Health and Nutrition Examination Survey (KNHANES). The bars from left to right are percentile groups (<5th percentile, underweight; 5th–85th percentile, normal-weight; 86th–94th percentile, overweight; and ≥95th percentile, obesity) of body mass index (BMI), waist circumference (WC), waist-to-height ratio (WHtR), or total-body fat mass (FM), separately for males and females. Especially, the gender- and age-specific percentile groups of BMI were stratified on the basis of the 2007 Korea National Growth Charts [38] (Appendix A). The percentile groups of WC, WHtR, and total-body FM were determined using the observed sample distributions in the study, separately for males and females (Appendix A). Results show mean ± 95% confidence interval of TBLH BMD for (**A**) males and (**B**) females stratified by the percentile groups of BMI, for (**C**) males and (**D**) females stratified by the percentile groups of WC, for (**E**) males and (**F**) females stratified by the percentile groups of WHtR, and for (**G**) males and (**H**) females stratified by the percentile groups of total-body FM. An analysis of covariance (ANCOVA) was used with age, menarcheal status (for females only), and total-body lean mass as covariates. Statistical significances were evaluated to test for whether there is a significant quadratic trend for the means of TBLH BMD to increase across the percentile groups of each obesity variable when controlling for the covariates. * The same letters indicate that the mean difference between groups is not significant (α = 0.05) on pairwise post-hoc comparisons using Bonferroni’s test.

**Table 1 jcm-12-05869-t001:** Characteristics of study population by gender in 2060 Korean adolescents aged 10 to 19 years from the KNHANES 2008–2011.

	Male (*n* = 1114)		Female (*n* = 946)		
Variable	Mean	(SD)	Mean	(SD)	*p*-Value *
Age (years)	14.17	(2.81)	14.28	(2.91)	0.372
Weight (kg)	56.91	(15.50)	50.23	(11.18)	<0.001
Height (cm)	163.78	(12.96)	157.21	(8.13)	<0.001
Obesity					
BMI (kg/m^2^)	20.88	(3.82)	20.28	(3.51)	<0.001
Waist circumference (cm)	71.09	(10.60)	67.23	(8.66)	<0.001
Waist-to-height ratio (cm/cm)	0.434	(0.057)	0.427	(0.049)	0.003
Body composition					
Lean mass (kg)	40.95	(10.73)	32.06	(5.62)	<0.001
Fat mass (kg)	13.42	(6.87)	16.27	(6.13)	<0.001
Bone mass (kg)	1.98	(0.58)	1.77	(0.39)	<0.001
Percent lean mass (%)	72.45	(7.56)	64.27	(5.64)	<0.001
Percent fat mass (%)	23.12	(8.08)	31.40	(5.92)	<0.001
Percent bone mass (%)	3.50	(0.53)	3.53	(0.49)	0.335
TBLH bone					
Bone mineral density (g/cm^2^)	0.887	(0.138)	0.834	(0.100)	<0.001
Bone mineral content (g)	1497.30	(490.56)	1268.94	(298.84)	<0.001
Bone area (cm^2^)	1646.94	(340.93)	1505.78	(233.89)	<0.001

KNHANES: Korea National Health and Nutrition Examination Survey; SD: standard deviation; BMI: body mass index; TBLH: total-body-less-head. * Statistical significances were assessed using *t*-test for two independent samples.

**Table 2 jcm-12-05869-t002:** Multiple regression results using total-body-less-head (TBLH) bone mineral density (BMD) as the dependent variable by gender in 2060 Korean adolescents aged 10 to 19 years from the KNHANES 2008–2011.

		Dependent Variable: TBLH BMD (g/cm^2^)							
		Male (*n* = 1114)				Female (*n* = 946)			
Parameter	Independent Variable	*b*	*p*-Value	Partial *R*	Ad-*R*^2^	*b*	*p*-Value	Partial *R*	Ad-*R*^2^
BMI	Intercept	0.4091	<0.001		0.736	0.3999	<0.001		0.608
	Age (years)	0.0080	<0.001	0.180		0.0060	<0.001	0.207	
	Menarcheal status *	NA				0.0396	<0.001	0.184	
	Lean mass (kg)	0.0096	<0.001	0.562		0.0099	<0.001	0.438	
	BMI (kg/m^2^)	−0.0014	0.072	−0.054		0.0000	0.990	0.000	
WC	Intercept	0.4092	<0.001		0.735	0.3903	<0.001		0.608
	Age (years)	0.0084	<0.001	0.194		0.0061	<0.001	0.209	
	Menarcheal status *	NA				0.0403	<0.001	0.187	
	Lean mass (kg)	0.0095	<0.001	0.563		0.0095	<0.001	0.422	
	WC (cm)	−0.0004	0.137	−0.045		0.0003	0.366	0.029	
WHtR	Intercept	0.4155	<0.001		0.735	0.3930	<0.001		0.608
	Age (years)	0.0083	<0.001	0.189		0.0061	<0.001	0.208	
	Menarcheal status *	NA				0.0402	<0.001	0.185	
	Lean mass (kg)	0.0093	<0.001	0.631		0.0097	<0.001	0.501	
	WHtR (cm/cm)	−0.0632	0.125	−0.046		0.0243	0.623	0.016	
Total-body FM	Intercept	0.4012	<0.001		0.738	0.3968	<0.001		0.608
	Age (years)	0.0073	<0.001	0.167		0.0061	<0.001	0.208	
	Menarcheal status *	NA				0.0395	<0.001	0.184	
	Lean mass (kg)	0.0098	<0.001	0.628		0.0102	<0.001	0.493	
	Total-body FM (kg)	−0.0013	<0.001	−0.109		−0.0004	0.436	−0.025	

KNHANES: Korea National Health and Nutrition Examination Survey; TBLH: total-body-less-head; BMD: bone mineral density; *b*: unstandardized beta coefficient; Partial *R*: partial correlation coefficient; Ad-*R*^2^; adjusted *R*^2^; BMI: body mass index; WC: waist circumference; WHtR: waist-to-height ratio; NA: not applicable. * Menarcheal status = 0 if pre-menarche, 1 if post-menarch.

**Table 3 jcm-12-05869-t003:** Multiple regression results representing curvilinear effects with the obesity variable squared (X^2^) as an independent variable by gender in 2060 Korean adolescents aged 10 to 19 years from the KNHANES 2008–2011.

		Dependent Variable: TBLH BMD (g/cm^2^)							
		Male (*n* = 1114)				Female (*n* = 946)			
Parameter	Independent Variable	*b*	*p*-Value	Partial *R*	Ad-*R*^2^	*b*	*p*-Value	Partial *R*	Ad-*R*^2^
BMI	Intercept	0.1306	0.013		0.743	0.1723	0.001		0.616
	Age (years)	0.0079	<0.001	0.180		0.0061	<0.001	0.211	
	Menarcheal status *	NA				0.0330	<0.001	0.153	
	Lean mass (kg)	0.0094	<0.001	0.556		0.0097	<0.001	0.436	
	BMI (kg/m^2^)	0.0251	<0.001	0.155		0.0219	<0.001	0.142	
	[BMI]^2^ (kg/m^2^)	−0.0006	<0.001	−0.166		−0.0005	<0.001	−0.144	
WC	Intercept	0.0011	0.988		0.743	−0.0635	0.456		0.621
	Age (years)	0.0080	<0.001	0.187		0.0059	<0.001	0.207	
	Menarcheal status *	NA				0.0329	<0.001	0.153	
	Lean mass (kg)	0.0093	<0.001	0.557		0.0095	<0.001	0.430	
	WC (cm)	0.0110	<0.001	0.160		0.0135	<0.001	0.181	
	[WC]^2^ (cm)	−0.0001	<0.001	−0.168		−0.0001	<0.001	−0.179	
WHtR	Intercept	0.0010	0.992		0.740	−0.0896	0.402		0.616
	Age (years)	0.0084	<0.001	0.194		0.0061	<0.001	0.212	
	Menarcheal status *	NA				0.0360	<0.001	0.167	
	Lean mass (kg)	0.0093	<0.001	0.631		0.0100	<0.001	0.514	
	WHtR (cm/cm)	1.7790	<0.001	0.125		2.1793	<0.001	0.149	
	[WHtR]^2^ (cm/cm)	−2.0057	<0.001	−0.130		−2.4083	<0.001	−0.148	
Total-body FM	Intercept	0.3752	<0.001		0.741	0.3536	<0.001		0.616
	Age (years)	0.0072	<0.001	0.167		0.0059	<0.001	0.206	
	Menarcheal status *	NA				0.0333	<0.001	0.155	
	Lean mass (kg)	0.0098	<0.001	0.629		0.0101	<0.001	0.494	
	Total-body FM (kg)	0.0027	0.021	0.069		0.0056	<0.001	0.128	
	[Total-body FM]^2^ (kg)	−0.0001	<0.001	−0.108		−0.0001	<0.001	−0.144	

KNHANES: Korea National Health and Nutrition Examination Survey; TBLH: total-body-less-head; BMD: bone mineral density; *b*: unstandardized beta coefficient; Partial *R*: partial correlation coefficient; Ad-*R*^2^; adjusted *R*^2^; BMI: body mass index; WC: waist circumference; WHtR: waist-to-height ratio; NA: not applicable. * Menarcheal status = 0 if pre-menarche, 1 if post-menarch.

## Data Availability

The data used in the study are accessible on the KNHANES website: https://knhanes,kdca.go.kr/knhanes/eng/index.go, accessed on 27 July 2023.

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
