# Peer review of "Inverted U-Shaped Relationship between Obesity Parameters and Bone Mineral Density in Korean Adolescents"

_jcm, 2023, doi:10.3390/jcm12185869_

Round 1
Reviewer 1 Report
General comments
This manuscript introduces and comprehensively investigates the impact of obesity on bone development in Korean adolescents aged 10–19 years. The topic is original and relevant to the field. There is limited information on this topic in the literature. There are no further improvements regarding the methodology. The conclusions are consistent with the evidence and arguments presented as well as summarize the main point of this article. Tables and figures are well formatted and make the study easy to follow.
Minor revision
Some references are not up-to-date (for example ref 1-3)
I would suggest adding some more recently published articles to the introduction section:
According to the World Health Organization, obesity is characterized as a chronic illness pertaining to adults, adolescents, and children worldwide, defined by a measured Body Mass Index (BMI) ≥ 30 kg/m2. It is estimated that 650 million adults are classified as obese worldwide, representing a staggering 13% of the adult population (1) In addition, obesity is a recognized risk factor for the development of comorbid conditions such as cardiovascular disease, type 2 diabetes mellitus, malignancy, asthma, osteoarthritis, chronic back pain, obstructive sleep apnoea, non-alcoholic fatty liver disease, and gallbladder diseases (2)
1) https://pubmed.ncbi.nlm.nih.gov/37568375/
2) https://pubmed.ncbi.nlm.nih.gov/35069068/
Reviewer 2 Report
Considering the fact that the global prevalence of obesity among children and adolescents has now reached alarming levels, defined by the World Health Organization (WHO) as a "global escalation of the epidemic", it is particularly important to ensure effective prevention not only of obesity, but also of related diseases, and these include disorders of bone mineralization.
The aim of the work was to try to answer the following questions:
1. Does obesity affect bone development in Korean adolescents aged 10–19?
2. Which indicators of obesity assessment are most correlated with skeletal disorders in young Koreans?
Currently, we are dealing with a global epidemic of obesity among children and adolescents. In 2019, the World Federation for Obesity estimated that in 2025, 206 million children and adolescents aged 5-19 will live with obesity, and in 2030, 254 million. In the presented study, the authors undertook to assess the impact of excessive body weight on the formation of bone structure disorders in the population of Korean children and adolescents. It is worth emphasizing the size of the study population amounting to as many as 2060 teenagers and including both girls/women and boys/men. Although studies on the impact of various environmental factors on the bone mineralization of children and adolescents are conducted in many countries, according to the reviewer, the work is original, as it presents the results of research on the population of young Koreans.
The strengths of the presented work include both its cognitive value and the possibility of using the obtained results to develop obesity prevention programs aimed at improving bone mineralization, taking into account the gender and age of Korean adolescents. The analysis of the results of the conducted studies showed a trend of an increase in BMD TBLH with an increase in the examined parameters of obesity (body mass index, waist circumference, waist circumference to height ratio and total adipose tissue mass); however, an inverted U-shaped pattern was observed, indicating that TBLH BMD decreased when obesity parameters reached excessive levels, which is a novelty. In other words, optimal BMD was achieved with moderate levels of obesity parameters, while both low and high levels of obesity parameters can lead to BMD deterioration. In addition, the advantage is the use of sampling in accordance with the KNHANES project, which ensures that the research sample is nationwide, which allows the generalization of the results to the Korean youth population.
The conclusions drawn by the authors are correct and result from the analysis of the research results obtained and the contemporary literature on the subject, and constitute an exhaustive answer to the
References are well chosen and correctly quoted in the text.
Tables presented in the publication are well constructed, legible and fully take into account the data contained in the content of the work.
Reviewer 3 Report
The chossen Thema from Authors for Artikle is very interesting with high scientific significance in the field of preventive public health policy.
Abstact include 221 words. In the part of introduction in section 4. on the end of introduction, my opinion to cancekate and also from tables parameter waist-to-heiight ratio, weil in publications is enough scientificallz to include BMI, Waist circumference, Total Fat, and percenteg e of fat mass. Insted od WHR only to include WC, waist circunference as a obesity indicator. In the part of Material and methods, regarding study population I didn t see the part of Ethical Approvment with sign number, and clinicalregistration for study. It is my opinion necesssery to include in the description of Material and Methods Ethical Approvment. The statistical data are excellent presentend, onlz mz suggestion in tableb 1. canceleted WHR, owith suggestion to stay WC. In part of discusion, suggestion to incude some 5 years up to date refrences, for example see below to amgliorate. Discusion is excellent written and pointed out importance of have in mind conections between obesity underweight, inflamation and bone mineral. Wihn the scope to prevent osteoporosis in adults, with analysis childhood and adolescent period, aimed to maintance helthz weight, with importance in public health policy Suggestions. the Authors must see the strategz to written refrence for JCM. I saw some errors, the article in references must be written in italic, year must be bold etc. example:
Chan, T.C.; Castillo, E.M.; Dunford, J.V.; Fisher, R.; Jensen, A.M.; Vilke, G.M.; Killeen, J.P. Hot spots and frequent fliers: Identifying high users of emergency medical services. Ann. Emerg. Med. 2012, 60, S83–S84
Suggestions for inclusion some 5 years up to date References.
Association between obesity and risk of fracture, bone mineral density and bone quality in adults: A systematic review and meta-analysis Anne-Frédérique Turcotte, Sarah O’Connor, Suzanne N. Morin, Jenna C. Gibbs, Bettina M. Willie, Sonia Jean, Claudia Gagnon PLoS One. 2021; 16(6): e0252487. Published online 2021 Jun 8. doi: 10.1371/journal.pone.0252487 PMCID: PMC8186797 ArticlePubReaderPDF–2.6MCite Select item 87449352. Bone Mineral Density Surveillance for Childhood, Adolescent, and Young Adult Cancer Survivors: Recommendations from the International Late Effects of Childhood Cancer Guideline Harmonization Group Jenneke E. van Atteveld, Renée L. Mulder, Prof. Marry M. van den Heuvel-Eibrink, Prof. Melissa M. Hudson, Prof. Leontien C.M. Kremer, Prof. Roderick Skinner, Prof. W. Hamish Wallace, Prof. Louis S. Constine, Claire E. Higham, Prof. Sue C. Kaste, Riitta Niinimäki, Sogol Mostoufi-Moab, Prof. Nathalie Alos, Danilo Fintini, Prof. Kimberly J. Templeton, Prof. Leanne M. Ward, Eva Frey, Roberto Franceschi, Vesna Pavasovic, Seth E. Karol, Nadia L. Amin, Lynda M. Vrooman, Prof. Arja Harila-Saari, Charlotte DemoorGoldschmidt, Robert D. Murray, Edit Bardi, Maarten H. Lequin, Prof. Maria Felicia Faienza, Olga Zaikova, Claire Berger, Stefano Mora, Prof. Kirsten K. Ness, Sebastian J.C.M.M. Neggers, Saskia M.F. Pluijm, Prof. Jill H. Simmons, Prof. Natascia Di Iorgi Lancet Diabetes Endocrinol. Author manuscript; available in PMC 2022 Sep 1. Published in final edited form as: Lancet Diabetes Endocrinol. 2021 Sep; 9(9): 622–637. Published online 2021 Jul 30. doi: 10.1016/S2213-8587(21)00173-X PMCID: PMC8744935 ArticlePubReaderPDF–3.1MCite Select item 87465183. Nutrition, Physical Activity, and Dietary Supplementation to Prevent Bone Mineral Density Loss: A Food Pyramid Mariangela Rondanelli, Milena Anna Faliva, Gaetan Claude Barrile, Alessandro Cavioni, Francesca Mansueto, Giuseppe Mazzola, Letizia Oberto, Zaira Patelli, Martina Pirola, Alice Tartara, Antonella Riva, Giovanna Petrangolini, Gabriella Peroni Nutrients. 2022 Jan; 14(1): 74. Published online 2021 Dec 24. doi: 10.3390/nu14010074 PMCID: PMC8746518 ArticlePubReaderPDF–1.8MCite Select item 93053864. New Insights on Bone Tissue and Structural Muscle-Bone Unit in Constitutional Thinness Mélina Bailly, Audrey Boscaro, Thierry Thomas, Léonard Féasson, Frédéric Costes, Bruno Pereira, Jorg Hager, Bruno Estour, Bogdan Galusca, Lore Metz, Daniel Courteix, David Thivel, Julien Verney, Natacha Germain Front Physiol. 2022; 13: 921351. Published online 2022 Jul 8. doi: 10.3389/fphys.2022.921351 PMCID: PMC9305386 ArticlePubReaderPDF–4.1MCite Select item 83604415. The association between overweight and obesity on bone mineral density in 12 to 15 years old adolescents in China Leishen Wang, Zhongxian Xu, Nan Li, Xuemei Meng, Shuo Wang, Chengshu Yu, Junhong Leng, Ming Zhao, Weiqin Li, Yanmei Deng Medicine (Baltimore) 2021 Aug 13; 100(32): e26872. Published online 2021 Aug 13. doi: 10.1097/MD.0000000000026872 PMCID: PMC8360441 ArticlePubReaderPDF–350KCite Select item 52252446. Body Composition and Bone Mineral Density in Patients with Heart Failure Demetrius A. Abshire, Debra, K. Moser, Jody L. Clasey, Misook L. Chung, Susan J. Pressler, Sandra B. Dunbar, Seongkum Heo, Terry A. Lennie West J Nurs Res. Author manuscript; available in PMC 2018 Jan 10. Published in final edited form as: West J Nurs Res. 2017 Apr; 39(4): 582–599. Published online 2016 Jul 11. doi: 10.1177/0193945916658885 PMCID: PMC5225244 ArticlePubReaderPDF–340KCite Select item 91022217. Secondary Osteoporosis and Metabolic Bone Diseases Mahmoud M. Sobh, Mohamed Abdalbary, Sherouk Elnagar, Eman Nagy, Nehal Elshabrawy, Mostafa Abdelsalam, Kamyar Asadipooya, Amr El-Husseini J Clin Med. 2022 May; 11(9): 2382. Published online 2022 Apr 24. doi: 10.3390/jcm11092382 PMCID: PMC9102221 ArticlePubReaderPDF–3.6MCite Select item 70643838. Effect of Aerobic or Resistance Exercise, or Both, on Bone Mineral Density and Bone Metabolism in Obese Older Adults While Dieting: A Randomized Controlled Trial Reina Armamento-Villareal, Lina Aguirre, Debra L Waters, Nicola Napoli, Clifford Qualls, Dennis T Villareal J Bone Miner Res. Author manuscript; available in PMC 2021 Mar 1. Published in final edited form as: J Bone Miner Res. 2020 Mar; 35(3): 430–439. Published online 2019 Dec 4. doi: 10.1002/jbmr.3905 PMCID: PMC7064383 ArticlePubReaderPDF–631KCite Select item 58799809. Association between pre‐sarcopenia, sarcopenia, and bone mineral density in patients with chronic hepatitis C Tatiana Bering, Kiara G.D. Diniz, Marta Paula P. Coelho, Diego A. Vieira, Maria Marta S. Soares, Adriana M. Kakehasi, Maria Isabel T.D. Correia, Rosângela Teixeira, Dulciene M.M. Queiroz, Gifone A. Rocha, Luciana D. Silva J Cachexia Sarcopenia Muscle. 2018 Apr; 9(2): 255–268. Published online 2018 Jan 19. doi: 10.1002/jcsm.12269 PMCID: PMC5879980 ArticlePubReaderPDF–443KCite Select item 734507910. Bone Mineral Density of Femur and Lumbar and the Relation between Fat Mass and Lean Mass of Adolescents: Based on Korea National Health and Nutrition Examination Survey (KNHNES) from 2008 to 2011 Aram Kim, Seunghui Baek, Seyeon Park, Jieun Shin Int J Environ Res Public Health. 2020 Jun; 17(12): 4471. Published online 2020 Jun 22. doi: 10.3390/ijerph17124471 PMCID: PMC7345079 ArticlePubReaderPDF–361KCite Select item 559217711. The Role of Overweight and Obesity on Bone Health in Korean Adolescents with a Focus on Lean and Fat Mass Hwa Young Kim, Hae Woon Jung, Hyunsook Hong, Jae Hyun Kim, Choong Ho Shin, Sei Won Yang, Young Ah Lee J Korean Med Sci. 2017 Oct; 32(10): 1633–1641. Published online 2017 Aug 30. doi: 10.3346/jkms.2017.32.10.1633 PMCID: PMC5592177 ArticlePubReaderPDF–1.0MCite Select item 604203812. Assessment of Biochemical Bone Turnover Markers and Bone Mineral Density in Thin and Normal-Weight Children Jadwiga Ambroszkiewicz, Joanna Gajewska, Grazyna Rowicka, Witold Klemarczyk, Magdalena Chelchowska Cartilage. 2018 Jul; 9(3): 255–262. Published online 2017 Jan 9. doi: 10.1177/1947603516686145 PMCID: PMC6042038 ArticlePubReaderPDF–252KCite Select item 661765513. Associations between body mass index, body composition and bone density in young adults: findings from a southern Brazilian cohort Isabel Oliveira Bierhals, Juliana dos Santos Vaz, Renata Moraes Bielemann, Christian Loret de Mola, Fernando Celso Barros, Helen Gonçalves, Fernando César Wehrmeister, Maria Cecília Formoso Assunção BMC Musculoskelet Disord. 2019; 20: 322. Published online 2019 Jul 9. doi: 10.1186/s12891-019-2656-3 PMCID: PMC6617655 ArticlePubReaderPDF–869KCite Select item 936824114. Obesity and Bone Health: A Complex Relationship Ana Piñar-Gutierrez, Cristina García-Fontana, Beatriz García-Fontana, Manuel Muñoz-Torres Int J Mol Sci. 2022 Aug; 23(15): 8303. Published online 2022 Jul 27. doi: 10.3390/ijms23158303 PMCID: PMC9368241 ArticlePubReaderPDF–1.0MCite Select item 613972615. How Is Adolescent Bone Mass and Density Influenced by Early Life Body Size and Growth? The Tromsø Study: Fit Futures—A Longitudinal Cohort Study From Norway Elin Evensen, Guri Skeie, Tom Wilsgaard, Tore Christoffersen, Elaine Dennison, Anne‐Sofie Furberg, Guri Grimnes, Anne Winther, Nina Emaus JBMR Plus. 2018 Sep; 2(5): 268–280. Published online 2018 Jun 7. doi: 10.1002/jbm4.10049 PMCID: PMC6139726 ArticlePubReaderPDF–284KCite Select item 902062816. The Association of Extreme Body Weight with Bone Mineral Density in Saudi Children Asmaa A. Milyani, Yousof O. Kabli, Abdulmoein E. Al-Agha Ann Afr Med. 2022 Jan-Mar; 21(1): 16–20. Published online 2022 Mar 18. doi: 10.4103/aam.aam_58_20 PMCID: PMC9020628 ArticlePubReaderCite Select item 286783317. Contributions of lean mass and fat mass to bone mineral density: a study in postmenopausal women Lan T Ho-Pham, Nguyen D Nguyen, Thai Q Lai, Tuan V Nguyen BMC Musculoskelet Disord. 2010; 11: 59. Published online 2010 Mar 26. doi: 10.1186/1471-2474-11-59 PMCID: PMC2867833 ArticlePubReaderPDF–1.7MCite Select item 945998318. Suboptimal Plasma Vitamin C Is Associated with Lower Bone Mineral Density in Young and Early Middle-Aged Men: A Retrospective Cross-Sectional Study Kuo-Mao Lan, Li-Kai Wang, Yao-Tsung Lin, Kuo-Chuan Hung, Li-Ching Wu, Chung-Han Ho, Chia-Yu Chang, Jen-Yin Chen Nutrients. 2022 Sep; 14(17): 3556. Published online 2022 Aug 29. doi: 10.3390/nu14173556 PMCID: PMC9459983 ArticlePubReaderPDF–481KCite Select item 877057119. Age-related changes in bone density, microarchitecture and strength in postmenopausal Black and White women: SWAN Longitudinal HR-pQCT Study Fjola Johannesdottir, Melissa S. Putman, Sherri-Ann M. Burnett-Bowie, Joel S. Finkelstein, Elaine W. Yu, Mary L. Bouxsein J Bone Miner Res. Author manuscript; available in PMC 2023 Jan 1. Published in final edited form as: J Bone Miner Res. 2022 Jan; 37(1): 41–51. Published online 2021 Nov 9. doi: 10.1002/jbmr.4460 PMCID: PMC8770571 ArticlePubReaderPDF–1.2MCite Select item 805664520. Influence of weight status on bone mineral content measured by DXA in children Francisco Sánchez Ferrer, Ernesto Cortes Castell, Francisco Carratalá Marco, Mercedes Juste Ruiz, José Antonio Quesada Rico, Ana Pilar Nso Roca BMC Pediatr. 2021; 21: 185. Published online 2021 Apr 20. doi: 10.1186/s12887-021-02665-5 PMCID: PMC8056645 ArticlePubReaderPDF–729KCite <<Author Response
Please see the attachment

Reviewer 4 Report
Thank you for inviting me to review this interesting manuscript by Lee et al titled
“Inverted U-shaped Relationship between Obesity Parameters and Bone Mineral Density in Korean Adolescents”. In general the manuscript is well written and the authors have presented an important concept relating the BMI and obesity to the bone mineral density in the developmental phase among 2,060, Korean adolescents aged 10–19 year. They have carried out rigorous statistical analysis of the results using multiple regression analysis, analysis of covariance and use of quadratic terms.
I only have a few minor comments.
The authors have mentioned that they observed age related changes. It will be interesting to know the number or percentage of participants that belonged to the different age groups.
It is generally understood that linear relationships are commonly used in literature as mentioned by the authors. It would be interesting to know why the authors thought of looking at a non-linear relationship and their choice for the statistical tool. It would also make it clearer to the reader if a brief description of how a second-order polynomial term improves the statistical analyses and in continuation of that what is the significance of using quadratic relationship in identifying a non-linear relationship and how it enhances the explanatory power of the models.
An important factor that stands out in the analysis is the number of participants in each of the percentile groups, mentioned in Supplementary Table 2 and Table3. The number in both the <5th and 95th are similar and lower compared to the other groups. Is it possible that this number influenced the results and provided the authors with the U shaped relationship?
The authors have dealt with many confounding variables and also adjusted for them. Will this relationship between the parameters still remain if the cut off of the percentiles is altered. How can they justify the results in terms of large variation in the number of participants. The authors have also mentioned that there was an age adjustment carried out but it will be interesting to know how old were the participants that fell in the extreme groups (<5th and 95th).
The statement ‘and understanding the factors that influence bone mineral accrual is essential for promoting optimal bone health’ is a little far-fetched as the only factor they studied is obesity. This can be rephrased.
The conclusion can be made more focused and directed. The word ‘excessive ‘and moderate can be changed to indicate the BMI category. Do consider if changing ‘obesity parameters’ to ‘measures of obesity’ suit the author’s manuscript.
Minor corrections are need.
Round 2
Reviewer 1 Report
Excellent work. All requested changes were addressed accordingly. It can be accepted for publication without further corrections